# Oxidation of *p*-[^125^I]Iodobenzoic Acid and *p*-[^211^At]Astatobenzoic Acid Derivatives and Evaluation In Vivo

**DOI:** 10.3390/ijms231810655

**Published:** 2022-09-13

**Authors:** Yawen Li, Ming-Kuan Chyan, Donald K. Hamlin, Holly Nguyen, Eva Corey, D. Scott Wilbur

**Affiliations:** 1Radiochemistry Division, Department of Radiation Oncology, University of Washington, Seattle, WA 98105, USA; 2GU Cancer Research Lab, Department of Urology, University of Washington, Seattle, WA 98195, USA

**Keywords:** astatine-211, iodine-125, radiolabel, oxidation, stability, biodistribution, astatate, iodate

## Abstract

The alpha particle-emitting radionuclide astatine-211 (^211^At) is of interest for targeted radiotherapy; however, low in vivo stability of many ^211^At-labeled cancer-targeting molecules has limited its potential. As an alternative labeling method, we evaluated whether a specific type of astatinated aryl compound that has the At atom in a higher oxidation state might be stable to in vivo deastatination. In the research effort, para-iodobenzoic acid methyl ester and dPEG_4_-amino acid methyl ester derivatives were prepared as HPLC standards. The corresponding para-stannylbenzoic acid derivatives were also prepared and labeled with ^125^I and ^211^At. Oxidization of the [^125^I]iodo- and [^211^At]astato-benzamidyl-dPEG_4_-acid methyl ester derivatives provided materials for in vivo evaluation. A biodistribution was conducted in mice with coinjected oxidized ^125^I- and ^211^At-labeled compounds. The oxidized radioiodinated derivative was stable to in vivo deiodination, but unfortunately the oxidized [^211^At]astatinated benzamide derivative was found to be unstable under the conditions of isolation by radio-HPLC (post animal injection). Another biodistribution study in mice evaluated the tissue concentrations of coinjected [^211^At]NaAtO_3_ and [^125^I]NaIO_3_. Comparison of the tissue concentrations of the isolated material from the oxidized [^211^At]benzamide derivative with those of [^211^At]astatate indicated the species obtained after isolation was likely [^211^At]astatate.

## 1. Introduction

There are only a few alpha-emitting radionuclides that have decay properties suitable for use in Targeted Alpha Therapy [1,2,3]. One radionuclide of high interest is the alpha particle-emitting radionuclide astatine-211 (^211^At). ^211^At is particularly attractive as it is a halogen, and in principle can be directly incorporated into biological targeting agents. Importantly, ^211^At is very efficient in killing cancer cells as it decays by a branched-chain process which effectively provides an alpha particle emission in 100% of the decays; furthermore, important is the fact that it does not have any alpha-emitting daughter nuclides after decay, which could cause toxicity in non-target tissues. While ^211^At is attractive for the development of Targeted Alpha Therapy agents, issues with in vivo stability of the label have been an impediment to the development of effective therapeutic agents [4,5].

Early studies demonstrated that ^211^At bonded to non-activated or deactivated aromatic ring pendant groups provided monoclonal antibody conjugates that are relatively stable towards in vivo deastatination [6,7,8]; however, it was found that this was not the case for more rapidly metabolized molecules. Thus, a number of investigators have evaluated chemical methods of stabilizing ^211^At on biomolecules to in vivo deastatination [4,9], but in most cases where there is a carbon-^211^At bond some release of ^211^At occurred in vivo. Interestingly, an example where the in vivo stability of the ^211^At is relatively high is in *meta*-[^211^At]astatobenzylguanidine ([^211^At]MABG) [10,11,12]. The reason for the higher in vivo stability of [^211^At]MABG is not known. Recently, a neopentyl glycol approach to labeling has shown promise for labeling with ^211^At [9], but this method may not be applicable to all biomolecules to be studied. We have investigated alternate ^211^At-bonding agents, such as anionic aryl-borates, which have been shown to provide an astatine-boron bond that is highly stable in vivo [13]; however, the conjugation of an anionic boron cage moiety to biomolecules can potentially affect their in vivo targeting properties, so for some molecules it might be best if the astatine could be incorporated directly into the organic molecule through a carbon-astatine bond. Thus, we have been interested in evaluating alternate ^211^At-carbon bonding approaches as a means of obtaining higher in vivo stability of the astatine bond.

In this investigation, we evaluated the effect on stability brought about by oxidizing an astatine atom that was bonded to a deactivated aryl-carbon atom. Since there are no stable isotopes of astatine, we used information on the stability of non-radioactive aryl-iodine molecules that have a higher oxidation state of the iodine atom [14,15]; it was noted that many aryliodine molecules with the iodine atom in a higher oxidation state are very effective organic oxidizing agents and thus would not be useful for in vivo applications; however, an example of a highly oxidized aryliodine, iodoxybenzoic acid, that was found to be quite stable in vitro seemed to warrant investigation of the in vivo stability of the corresponding radioiodinated and astatinated compounds [16]. Thus, we synthesized some iodo-, radioiodo- and astatobenzoic acid derivatives, then oxidized the bonded (radio)halogen, isolated the products from that oxidation process, and evaluated biodistributions in mice of the isolated products in mice. The results of that investigation are described herein.

## 2. Results

### 2.1. Syntheses of Compounds 

The compounds synthesized for this investigation are shown in Figure 1. A literature report indicated that *p*-iodoxybenzoic acid, **2**, was stable in vitro [16], so that was our first synthetic target. An early report of the preparation of *p*-iodoxybenzoic acid involved the preparation of *p*-iodosobenzoic acid (also named as *p*-iodosybenzoic acid), which disproportionates to *p*-iodobenzoic acid, **1**, and *p*-iodoxybenzoic acid, **2** [16]; however, that approach was not viewed as efficient, so a more direct synthesis was sought; it had been reported that iodoxybenzene derivatives could be prepared using refluxing aqueous sodium iodate, but under the conditions used *p*-iodosylbenzoic acid was obtained rather than **2** [17]. Another approach to the synthesis of iodoxybenzene derivatives from iodobenzene derivatives was the use of oxone as the oxidant [18,19]. 

In this study, we investigated the use of K_2_S_2_O_8_, NaIO_4_ and *m*-chloroperbenzoic acid (mCPBA) as oxidizing agents in solvent mixtures of MeOH/H_2_O, DMF/H_2_O or DMSO/H_2_O, heating from 60 °C to 150 °C in the microwave. The studies began with oxidation of (non-radioactive) *para*-iodobenzoic acid, **1** (Figure 1), to prepare the iodoxybenzoic acid. Initial oxidation studies to prepare the **2** by reaction of **1** with NaIO_4_ or K_2_S_2_O_8_ resulted in mixtures of products, none of which could be identified as the desired *para*-iodoxybenzoic acid, **2**. Interestingly, it was found that HCl could not be used in the reaction as that resulted in the formation of *para*-chlorobenzoic acid under the reaction conditions studied; it appeared that the carboxylate interferes with the oxidation reaction, so the methyl ester, **3**, was prepared and used in subsequent studies. Further studies demonstrated that the use of *m*CPBA provided the highest yields under the conditions studied, so those reactions are described.

Oxidation of benzoic acid methyl ester **3** with *m*CPBA under microwave heating at 70 °C for 20 min provided *para*-iodoxybenzoic acid methyl ester **6** to use as an HPLC standard. Compound **6** had the correct exact mass by HRMS analysis and the NMR spectra showed consistent changes in the aryl proton shifts for converting **5** [δ 7.75 (d), 7.81 (d)] to **6** [δ 8.09 (d), 8.13(d)], but it was very insoluble in all solvents tried. Despite the low solubility of the non-radioactive **6**, studies were conducted to determine if radioiodinated and astatinated benzoic acid methyl esters could be used to prepare the corresponding [^125^I]iodoxy- and [^211^At]astatoxybenzoic acid methyl esters, [^125^I]**6** and [^211^At]**8** respectively. The reactions conducted to prepare oxidized aryl radiohalogens [^125^I]**6** and [^211^At]**8** are shown in Figure 2.

It was apparent that the oxidized (radio)halobenzoic acid derivatives needed to be more soluble in aqueous solutions to be of value in the development of new radiopharmaceuticals; this fact led to the synthesis of amino-dPEG_4_-carboxylic acid, **16**, adducts of *para*-tri-*n*-butylstannylbenzoic acid and *para*-iodobenzoic acid, to form *para*-substituted benzamidyl-dPEG_4_-carboxylate methyl esters **9** and **10** respectively, as shown in Figure 3.

The synthesis of the arylstannane derivative **9**, which can be used for radiolabeling, began by reaction of *para*-iodobenzoic acid, **1**, with trifluoroacetic acid tetrafluorophenyl ester (TFA-OTFP) to prepare the *para*-iodobenzoic acid tetrafluorophenyl (TFP) ester **14** [20]. In the next step, the *para*-tri-*n*-butylstannylbenzoic acid TFP ester **15** was prepared using a palladium catalyzed stannylation reaction [20]. Preparation of the amino-dPEG_4_-acid adduct, **17a**, was accomplished by reaction of **15** with amino-dPEG_4_-acid, **16**, and Et_3_N in DMF at room temperature. The *para*-tri-*n*-butylstannylbenzamidyl-dPEG_4_-carboxylic acid TFP ester, **18a**, was prepared by reaction of TFA-OTFP with **17a** and Et_3_N in CHCl_3_ at room temperature. Conversion of **18a** to the methyl ester **9** was accomplished by reaction in anhydrous methanol and dimethylaminopyridine (DMAP).

Non-radioactive *para*-iodobenzamidyl-dPEG_4_-carboxylic acid methyl ester, **10** was prepared as an HPLC standard. In that synthesis **17b** was prepared by reaction of **14** with **16** in DMF containing Et_3_N at room temperature. Compound **17b** was subsequently esterified using TFA-OTFP to form the TFP ester **18b**. Following that **18b** was converted to the methyl ester **10** by reaction with anhydrous methanol and triethylamine Et_3_N.

### 2.2. Radiohalogen Labeling and Oxidation of Radioiodine and Astatine Compounds

Reactions of tri-*n*-butylstannylbenzoic acid methyl ester, **4** (Figure 2), with [^125^I]NaI or [^211^At]NaAt in the presence of chloramine-T (ChT) resulted in 100% and 83% radiochemical yields (RCY) of the radiohalogenated products [^125^I]**5** and [^211^At]**7** respectively. Similarly, radiohalogenation of *para*-tri-*n*-butylstannylbenzamidyl-dPEG_4_-carboxylic acid methyl ester **9** (Figure 3) using ChT as oxidant provided quantitative yields of [^125^I]**10** and [^211^At]**12** by radio-HPLC analyses (Figure 4A,B). The radiohalogenation reaction products were used without purification in the oxidation reactions to make the corresponding iodoxyaryl or astatoxyaryl compounds. Oxidation of [^125^I]**5** (Figure 2) provided [^125^I]**6** in 71% RCY. Oxidation of [^125^I]**10** (Figure 3) at 70 °C for 20 min provided a complex mixture of products, but isolation of the major peak (~6–8 min) provided [^125^I]**11** in 60% RCY (Figure 4C,E). Retention time of the isolated [^125^I]**11** was consistent with the non-radioactive standard on the radio-HPLC. Oxidation of the astatobenzoate methyl ester [^211^At]**7** (t_R_ = 14.8 min) at 50 °C provided a new compound, presumably [^211^At]**8** in 83% RCY by radio-HPLC analysis, t_R_ = 5.6 min. Similarly, oxidation of [^211^At]**12** (t_R_ = 12.7 min) at 50 °C provided a new compound by radio-HPLC analyses, presumably [^211^At]**13** in 74% RCY, t_R_ = 5.8 min. The HPLC retention times seen for the oxidized compounds [^211^At]**8** and [^211^At]**13** had similar shifts to that observed in the conversion of **5** (t_R_ = 14.4 min) to **6** (t_R_ = 3.4 min) and **12** (t_R_ = 12.5 min) to **13** (t_R_ = 5.8 min). Radiolabeled [^125^I]**11** and [^211^At]**13** were isolated from the HPLC effluent, evaporated and redissolved in phosphate buffered saline (PBS) for evaluation of tissue distributions in athymic mice.

Preparation of [^125^I]NaIO_3_ and [^211^At]NaAtO_3_ was of interest for the studies. Initial oxidation studies were conducted to prepare [^125^I]NaIO_3_ from commercially available [^125^I]NaI using NaI and NaIO_3_ as HPLC standards. Although there are a number of oxidants that might be used, it was found that oxidation of [^125^I]NaI using NaIO_4_ in dilute sulfuric acid provided quantitative conversion to [^125^I]NaIO_3_. Oxidation of [^211^At]NaAt provided a new species, presumably [^211^At]NaAtO_3_ using the same reagents and conditions. Interestingly, the oxidation of [^125^I]NaI required 3 days for complete conversion, whereas it only took 30 min for oxidation of [^211^At]NaAt under the conditions used.

### 2.3. Biodistribution Studies

Two biodistribution studies were conducted. One biodistribution study evaluated the tissue distribution in athymic mice of isolated products from the reactions to produce [^125^I]**11** and [^211^At]**13**. Isolation of [^125^I]**11** from the reaction mixture and reinjection after evaporation of solvents and dissolution in PBS showed that it did not change during that process (Figure 4E). In contrast, the product obtained by isolation of the radiochromatogram peak at a retention time of 5.8 min (Figure 4D), believed to be [^211^At]**13**, had a major peak at 4.3 min and a minor peak at 3.4 min (Figure 4F). Unfortunately, analyses of the isolated solution containing [^211^At]**13** was conducted after coinjection of [^125^I]**11** and [^211^At]**13** into mice due to timing required to conduct the biodistribution studies. The tissue concentrations of ^125^I and ^211^At at 1, 4 and 24 h post injection (p.i.) are plotted in Figure 5 (tabulation of data in Appendix A). The fact that the concentrations (%ID/g) of ^125^I in stomach and neck tissues are low is an indicator that iodoxyaryl compounds are stable to in vivo deiodination. In contrast, the tissue concentrations obtained for the isolated (decomposed) [^211^At]**13** are similar to free ^211^At [13], suggesting that an ^211^At species was injected, most likely a mixture of [^211^At]astatide and [^211^At]astatate based on the oxidative conditions used prior to isolation and the radio-HPLC retention times observed (Figure 4F).

Based on the results from the initial biodistribution study, a comparison with the tissue distribution of [^211^At]astatate was of interest. A second biodistribution study evaluated the tissue concentrations of [^125^I]iodate and [^211^At]astatate in athymic mice. After the conversion of [^125^I]NaI and [^211^At]NaAt to [^125^I]NaIO_3_ and [^211^At]NaAtO_3_, (Figure 6A,B) the reaction solutions were isolated from the HPLC effluent to remove oxidant and solvents were evaporated with heating. The residues remaining were dissolved in PBS and re-examined by radio-HPLC. Radiochromatograms of the isolated products are shown in Figure 6C,D. Under the conditions of isolation (evaporation with heating) mixtures of compounds were obtained for both the [^125^I]NaIO_3_ and [^211^At]NaAtO_3_. Thus, in a second preparation of [^125^I]NaIO_3_ and [^211^At]NaAtO_3_, the HPLC solvent was removed without heating (room temperature), followed by dissolution in PBS; this provided [^125^I]NaIO_3_ and [^211^At]NaAtO_3_ in higher purity (Figure 6E,F), which allowed them to be used directly in the biodistribution study. The biodistribution study was conducted under the same conditions as the coinjected [^125^I]**11** and [^211^At]**13** study. The tissue concentrations of ^125^I and ^211^At at 1, 4 and 24 h post-injection (p.i.) are plotted in Figure 7 (tabulation of data in Appendix A). As expected, the concentrations of [^125^I]IO_3_^–^ and [^211^At]AtO_3_^–^ in the neck (including thyroid) and stomach are significantly higher than those in the other tissues. The concentration of [^125^I]IO_3_^–^ in the lung, spleen and kidney is similar to that in other tissues, whereas, the concentration of [^211^At]AtO_3_^–^ in the lung (7.1 %ID/g @ 1 h p.i., 5.8 %ID/g @ 4 h p.i.) and spleen (6.8 %ID/g @ 1 h p.i., 6.4 %ID/g @ 4 h p.i.) is higher than that in the liver, intestine and muscle (<1.6 %ID/g @ 1 and 4 h p.i.). Essentially, all [^125^I]IO_3_^–^ activity cleared out from the body by 24 h p.i., but a small percentage of injected ^211^At activity remained in the neck (4.3 %ID/g), stomach (2.0 %ID/g) and lung (1.0 %ID/g) at 24 h p.i.

## 3. Discussion

The primary goal of this study was to evaluate whether a specific type of astatinated aryl compound that has the At atom in a higher oxidation state might be stable to in vivo deastatination. We were aware that iodoaromatic compounds which have the iodine atom in a higher oxidation state were generally unstable due to their propensity to oxidize other organic molecules [15,21], but a specific example of a *para*-iodoxybenzoic acid was found to be quite stable in vitro [16]. Thus, we chose to use *para*-[^125^I]iodoxylbenzoate and *para*-[^211^At]astatoxybenzoate derivatives as model compounds to evaluate their in vivo stability; these model compounds were attractive as this approach could potentially be readily adapted to many astatinated and radioiodinated small molecules.

Initial studies were conducted to prepare *p*-(radio)haloxybenzoic acids and their methyl esters as simple model compounds. While the benzoic acid methyl esters could be prepared, radiolabeled and oxidized as desired, those compounds were found to be quite insoluble in all solvents tried; this led to conjugation of a linker molecule, amino-dPEG_4_-carboxylic acid, **16**, with *para*-iodobenzoic acid and *para*-tri-*n*-butylstannylbenzoic acid esters to increase the aqueous solubility of the compounds to be tested. Preparation of the stannylbenzoate derivatives was conducted as intermediates for the radiohalogenation reactions [22,23,24]. The *para*-(radio)halobenzoic acid derivatives containing the amino-dPEG_4_ acid linker had improved solubility and the corresponding radioiodinated and astatinated derivatives [^125^I]**10** and [^211^At]**12**, could be readily prepared from the stannyl intermediate **9**. Subsequent oxidation with *m*CPBA provided the *para*-[^125^I]iodoxybenzamide derivative [^125^I]**11** based on its radio-HPLC retention time, which corresponded with the NMR and MS characterized non-radioactive standard **10**. Contrary to that, the radio-HPLC retention time of the astatinated molecule [^211^At]**13** was significantly different from that of the radioiodinated product, making it more difficult to be certain of the identity of the product; however, the change in retention time for the astatinated compounds was consistent with those of the radioiodinated compounds and consistent with other compounds studied in our laboratory. Thus, it seems very likely that the astatinated compounds in Figure 3 have the structures shown.

The issue addressed in this study was stability of astatinated and radioiodinated aromatic compounds when injected into living animals. Nature provides deiodinase enzymes to recycle iodine for thyroid hormones [25] and also provides enzymes that dehalogenate compounds to detoxify them [26,27,28]. There are several different mechanisms that are used by enzymes to dehalogenate compounds, including hydrolytic substitution reactions and reductive or oxidative elimination reactions [26]. The family of deiodinase enzymes removes specific iodine atoms from thyroid hormones through a reductive elimination of iodine from the aromatic ring [29]. While the deiodinase enzymes may play a role in deiodination of some radioiodinated phenolic compounds [30], radioiodination of “deactivated” aromatic compounds (e.g., benzoates) has proven to stabilize them from in vivo deiodination; these non-activated aromatic ring compounds have been used to help stabilize carbon-astatine bonds [31,32], but many rapidly metabolized ^211^At-labeled compounds have been found to be susceptible to deastatination [4]; it has been shown that oxidative dehalogenation can induce loss of the ^211^At atom from astatobenzoates [33]. In the same study, quantum mechanical calculations indicated that astatobenzoates could undergo oxidative deastatination some 6 million times faster at 37 °C than the corresponding iodobenzoates.

The concept that oxidative deastatination as a mechanism for releasing ^211^At in vivo led to a hypothesis that greater stability might be obtained if the astatine atom in arylastatine is stable in a higher oxidation state and cannot be further oxidized by the enzymatic oxidants in cellular compartments (e.g., lysosomes). In this study we found that oxidation of the At atom provided a new compound (presumably) the arylastatoxy compound, but that compound was unstable to the conditions used for isolation. While this result might seem to warrant further studies to determine if that compound could be isolated without decomposition, it was thought that the low stability of the compound would ultimately release the At in vivo, so no further studies were conducted.

## 4. Materials and Methods

### 4.1. General

*Caution must be taken when working with aryliodine compounds where the iodine is oxidized to the + 5 oxidation state* as some derivatives have been reported as explosive under excessive heating or impact [34]; however, it is thought that the explosive properties may have been from bromate impurities when potassium bromate in sulfuric acid was used to prepare the aryliodine (V) compound.

All reagents used were obtained from commercial sources as analytical grade or better and were used without further purification. *meta*-Chloroperbenzoic acid, chloramine-T, bis-tributyltin, and tetrakis (triphenylphosphine)palladium [Pd(Ph_3_P)_4_] were obtained from Sigma-Aldrich (St. Louis, MO, USA). *p*-Iodobenzoic acid, **1**, and its methyl ester **3** were obtained from commercial sources (i.e., Sigma-Aldrich). Amino-dPEG_4_-acid, **16**, was provided as a gift from Quanta BioDesign (Plain City, OH, USA). 2,3,5,6-Tetrafluorophenyl trifluoroacetate (TFA-OTFP) was prepared as previously described [35]. Microwave reactions were conducted in a Biotage Initiator 2.0 using 0.5–2 mL vials. Samples were taken from the vial for HPLC analyses by puncturing the vial septum with a needle and withdrawing 10 μL. Reaction progress could be evaluated at multiple time points using this method.

### 4.2. Radioactive Materials

All radioactive materials were used under a UW Radiation Use Authorization and handled according to approved Radiation Safety protocols at the University of Washington. [^125^I]NaI was purchased from Perkin-Elmer Life and Analytical Sciences, Inc. (Waltham, MA, USA), as high concentration solutions in 0.1 N NaOH. Radioiodinations and astatinations were conducted in a charcoal-filtered Plexiglas enclosure (Biodex Medical Systems, Inc., Shirley, NY, USA) housed in a radiochemical fume hood. Astatine-211 (^211^At) was produced in-house at the University of Washington Medical Cyclotron Facility. ^211^At was produced by cyclotron irradiation via the ^209^Bi(α,2n)^211^At reaction [36] and was isolated from the irradiated Bi targets using a “wet chemistry” method as described previously [37]. Quantification of ^125^I was accomplished on a Capintec CRC-15R Radioisotope Calibrator using the manufacturer’s recommended calibration number. Quantification of ^211^At was accomplished on a Capintec CRC-15R and a Capintec 55tR using cross-calibration against a High Purify Germanium detector as described previously [37].

### 4.3. Chromatography Equipment and Conditions

Non-radioactive compounds were purified using a Biotage Flash Purification System (Charlottesville, VA, USA). The purifications were conducted on a reversed-phase C18 FLASH 25 + M column. The column was eluted by a gradient mixture composed of MeOH and 0.1% aqueous acetic acid, pH 3.25. Starting with 25% MeOH, the initial solvent mixture was held for 2 min, increased to 100% MeOH over the next 12 min, and then held at 100% MeOH for 8 min. Fractions were collected based on UV detection at 254 nm. Identification of fractions containing the desired product was achieved using analytical HPLC. The fractions containing pure compound were combined, the solvent was evaporated, and the product was isolated to provide the yields listed.

HPLC analyses of non-radioactive compounds were performed using a system composed of a Hewlett-Packard quaternary 1050 gradient pump, a variable wavelength UW detector (254 nm) and an evaporative light scattering detector (ELSD 2000, Alltech, Deerfield, IL, USA). Radio-HPLC analysis of product mixtures from radiohalogenation reactions were analyzed using a system composed of two Beckman model 110 B pumps, a Beckman 420 controller, a Beckman model 153 UV detector (254 nm), and a Beckman model 170 radiation detector. Analysis of the HPLC data was conducted on Hewlett-Packard HPLC ChemStation software. Reversed-phase HPLC analyses were carried out on an Alltech Altima C18 column (5 μm, 250 × 4.6 mm) using a gradient solvent system at a flow rate of 1 mL/min. The gradient mixture was composed of MeOH and 0.1% aqueous HOAc (pH 3.25). The gradient started with 40% MeOH for 2 min and increased linearly to 100% MeOH over 10 min, followed by elution with 100% MeOH for 8 min. The retention time (t_R_) of each compound is provided with the experimental procedure. Radio-HPLC analyses of ^125^I, ^211^At, and stable iodine ions were conducted by using a JMC J’Sphere ODS M80 column (5 μm, 250 × 4.6 mm) eluting isocratically at 1 mL/min with a 1:1 mixture of acetonitrile (ACN):50 mM aqueous tetrabutylammonium dihydrogen phosphate (pH 4.5).

### 4.4. Spectral Analyses

All new non-radioactive compounds were characterized using ^1^H NMR and high resolution mass spectral (HRMS) analyses. The spectral data obtained is provided with the synthesis procedures. ^1^H NMR spectra were collected on a Bruker AV-500 (500 MHz for ^1^H) spectrometer, referencing to tetramethylsilane as an internal standard (δ = 0.0 ppm). HRMS data were obtained on a Bruker Apex-Qe FT/ICR instrument. The elemental composition calculations were conducted using the Bruker software (DataAnalysis 3.4).

### 4.5. Compound Syntheses and Radiolabeling

#### 4.5.1. *Para*-Tri-*n*-Butylstannylbenzoic Acid Methyl Ester, **4**

*Para*-Tri-*n*-butylstannylbenzoic acid methyl ester **4** was prepared by reaction of *para*-iodobenzoate methyl ester, **3** (Sigma-Aldrich) with bis(tribuyltin) and tetrakis(triphenylphosphine)-palladium [Pd(Ph_3_P)_4_] as previously described [20]. HPLC t_R_ = 22.8 min. HRMS (ES^+^): C_20_H_35_O_2_Sn (M+H)^+^ Calcd: 427.1654. Found: 427.1660.

#### 4.5.2. *Para*-[^125^I]Iodobenzoic Acid Methyl Ester, [^125^I]**5**

Radioiodination of **4** was achieved by adding 10 μL of a 1 mg/mL aqueous solution of ChT to a 100 μL solution of 0.05 N HCl MeOH/H_2_O containing 37 MBq (1 mCi) of [^125^I]NaI and 100 μg of ester **4**. The mixture sat at room temperature for 5 min before conducting radio-HPLC analysis. The reaction mixture was analyzed by radio-HPLC for preparation of [^125^I]**5** and elution with the non-radioactive HPLC standard, **3**. Radio-HPLC indicated that a quantitative conversion was obtained.

#### 4.5.3. *Para*-Iodoxybenzoic Acid Methyl Ester, **6**

A solution containing 0.5 g (1.91 mmol) of **3** and 1.45 g (4.20 mmol; 50% purity) of *m*CPBA in 18 mL of MeOH was stirred and heated under microwave heating at 70 °C for 20 min. A solid precipitated from the MeOH solution. The reaction mixture was filtered, the residue was washed with MeOH (3 × 20 mL), dried in the open air to obtain a 58 mg (10.3%) yield of **6** as a white solid, m.p. 243–245 °C. The product was only partially soluble in DMSO at room temperature. HPLC t_R_ = 3.4 min. ^1^H NMR (DMSO-d_6_): 3.89 (s, 3H), 8.09 (d, 2H, Ar*H*); 8.13 (d, 2H, Ar*H*). HRMS (ES^+^): C_8_H_8_IO_4_ (M+H)^+^ Calcd: 294.9467 Found: 294.9473.

#### 4.5.4. *Para*-[^125^I]Iodoxybenzoate Methyl Ester, [^125^I]**6**

To the reaction mixture containing [^125^I]**5** (Section 4.5.2. above) was added 100 μL of a 0.05 M solution of *m*CPBA in MeOH. The solution was stirred and heated at 70 °C for 20 min using microwave heating. After that time, the reaction solution was analyzed by radio-HPLC to show that [^125^I]**6** was formed in 71% radiochemical yield.

#### 4.5.5. *Para*-[^211^At]Astatobenzoic Acid Methyl Ester, [^211^At]**7**

Astatination of **4** was achieved by adding 20 μL of a 1 mg/mL aqueous solution of ChT to a 100 μL solution of 0.05 N HCl MeOH/H_2_O containing 70 MBq (1.9 mCi) of [^211^At]NaAt and 100 μg of ester **4**. The mixture sat at room temperature for 5 min before conducting radio-HPLC analysis. The reaction mixture was analyzed by radio-HPLC for preparation of [^211^At]**7**. Radio-HPLC indicated that an 83% radiochemical yield of [^211^At]**7** was obtained.

#### 4.5.6. *Para*-[^211^At]Astatoxybenzoic Acid Methyl Ester, [^211^At]**8**

To the reaction mixture containing [^211^At]**7** (Section 4.5.5. above) was added 100 μL of a 0.05 M solution of *m*CPBA in NaOH. The solution was stirred and heated for 20 min under microwave heating at 50 °C. After that time the reaction solution was analyzed by radio-HPLC, which indicated a 58% radiochemical yield of [^211^At]**8** was obtained.

#### 4.5.7. *Para*-Iodobenzoic Acid TFP Ester, **14**

*Para*-Iodobenzoic acid TFP ester, **14**, was synthesized from *4*-iodobenzoic acid, **1**, by reaction with TFA-OTFP and triethylamine (Et_3_N) as previously described [13].

#### 4.5.8. *Para*-Tri-*n*-Butylstannylbenzoic Acid TFP Ester, **15**

*Para*-Tri-*n*-butylstannylbenzoic acid TFP ester, **15**, was prepared in 83% yield by reaction of **14** with bis-tributyltin, (Bu_3_Sn)_2_, and tetrakis(triphenylphosphine)palladium [Pd(Ph_3_P)_4_] as previously described [13].

#### 4.5.9. *Para*-Tri-*n*-Butylstannylbenzamidyl-dPEG_4_-Carboxylic Acid, **17a**

Intermediate **17a** was synthesized by adding H_2_N-dPEG_4_-CO_2_H, **16** (0.522 g, 1.967 mmol) to a solution of **15** (1 g, 1.788 mmol), Et_3_N (0.498 mL, 3.58 mmol), CH_2_Cl_2_ (20 mL), and anhydrous DMF (5 mL). The mixture was stirred at room temperature for 4 h. After that period volatile materials were removed by rotary evaporator under vacuum, then the crude product was purified by silica gel column (2.5 cm × 25 cm) eluting with a gradient solution from 100% EtOAc to 50:50 MeOH:EtOAc to yield 61% of **17a** as a colorless tacky solid. HRMS (ES^+^) calcd for C_30_H_53_NNaO_7_Sn (M+Na)^+^: 682.2736. Found: 682.2725. HPLC: t_R_ = 16.8 min.

#### 4.5.10. *Para*-Iodobenzamidyl-dPEG_4_-Carboxylic Acid, **17b**

Intermediate **17b** was synthesized by reaction of H_2_N-dPEG_4_-CO_2_H, **16** (0.638 g, 2.41 mmol) to a solution of **14** (1.0 g, 2.52 mmol), Et_3_N (0.50 mL, 3.6 mmol), and anhydrous DMF (15 mL). The mixture was stirred at room temperature for 30 min. After volatile materials were evaporated by rotary evaporator under vacuum, the crude product was purified by silica gel column (1.5 cm × 25 cm) eluted with gradient solution from 100% EtOAc to 30% MeOH:EtOAc to yield 0.98 g (82%) of **17b** as a colorless tacky solid. ^1^H NMR (CDCl3): 2.7 (t, 2H, J = 6.5 Hz), 2.80 (t, 2H, J = 5.7 Hz), 3.66–3.70 (m, 14H), 3.76 (t, 2H, J = 6.4Hz), 7.08 (s, 1H), 7.59 (d, 2H, J = 8.8 Hz), 7.80 (d, 2H, J = 8.8 Hz). HRMS (ES+) calcd for C_18_H_27_INO_7_ (M+H)^+^ 496.0832. Found: 496.0831. HPLC: t_R_ = 11.6 min.

#### 4.5.11. *Para*-Tri-*n*-Butylstannylbenzamidyl-dPEG_4_-Carboxylic Acid TFP Ester, **18a**

To a solution containing 1 g (0.98 mmol) of **17a** and 0.206 mL (1.48 mmol) Et_3_N in 15 mL of anhydrous CHCl_3_ was added 0.204 mL (1.185 mmol) of TFA-OTFP. The solution was stirred at room temperature for 30 min to prepare the TFP ester **18a**. The solution containing **18a** was used in the next step without purification.

#### 4.5.12. *Para*-Tri-*n*-Butylstannylbenzamidyl-dPEG_4_-Carboxylic Acid Methyl Ester, **9**

To the crude solution of **18a** was added 1 mL of anhydrous methanol and 0.121 g (0.987 mmol) of DMAP. The resultant solution was stirred at room temperature for another 2 h; then, the volatile materials were removed using a rotary evaporator under vacuum. The crude **9** was purified by silica gel column (2.5 cm × 25 cm) eluting with a gradient beginning with 1:1 ethyl EtOAc:hexanes to 100% EtOAc, then to 10% MeOH/EtOAc and finally to 20% MeOH:EtOAc. Compound **9** was isolated in 86% yield as a colorless oil. HRMS (ES^+^) calcd for C_31_H_56_NO_7_Sn (M+H)^+^: 674.3073. Found: 674.3097. HPLC: t_R_ = 17.7 min.

#### 4.5.13. *Para*-Iodobenzamidyl-dPEG_4_-Carboxyic Acid TFP Ester, **18b**

To a solution containing 1 g (2.08 mmol) of **17b** and 0.434 mL (3.12 mmol) Et_3_N in 15 mL anhydrous CHCl_3_ was added 0.43 mL (2.49 mmol) TFA-OTFP. That solution was stirred at room temperature for 5 min then analyzed by HPLC. The HPLC confirmed that the TFP ester, **18b**, was formed quantitatively. The solution containing **18b** was used in the next step without purification.

#### 4.5.14. *Para*-Iodobenzamidyl-dPEG_4_-Carboxyic Acid Methyl Ester, **10**

To the crude solution of **18b** was added 2 mL of anhydrous MeOH, and the resultant solution was stirred at room temperature for another hour. Volatile materials were removed by rotary evaporation under vacuum. The crude **10** was purified by silica gel column (1.5 cm × 25 cm) eluting with a gradient solution from 40% EtOAc/hexanes to 10% MeOH/EtOAc. The methyl ester **10** was isolated in 92% yield as a colorless tacky solid. ^1^H NMR (CDCl_3_): 2.6 (t, 2H, J = 6.5 Hz), 2.79 (t, 2H, J = 5.7 Hz), 3.61 (s, 3H), 3.64–3.69 (m, 14H), 3.75 (t, 2H, J = 6.4Hz), 7.06 (s, 1H), 7.59 (d, 2H, J = 8.8 Hz), 7.80 (d, 2H, J = 8.8 Hz). HRMS (ES^+^) calcd for C_19_H_28_INNaO_7_ (M+Na)^+^: 532.0803. Found: 532.0800. HPLC: t_R_ = 12.5 min.

#### 4.5.15. *Para*-[^125^I]Iodobenzamidyl-dPEG_4_-Carboxylic Acid Methyl Ester, [^125^I]**10**

To a solution containing 100 μg **9** in 100 μL of a 0.05 N HCl MeOH/H_2_O solution and 2 μL of a [^125^I]NaI solution (~37 MBq;1 mCi) was added 10 μL of a 1 mg/mL aqueous solution of ChT. The mixture sat at room temperature for 5 min before conducting radio-HPLC analysis for the preparation of [^125^I]**10**. Radio-HPLC indicated that ~100% conversion was obtained. Radio-HPLC t_R_ = 12.7 min.

#### 4.5.16. *Para*-[^211^At]Astatobenzamidyl-dPEG_4_-Carboxylic Acid Methyl Ester, [^211^At]**12**

To a solution containing 100 μg **9** in 100 μL of a 0.05 N HCl MeOH/H_2_O and 200 μL of a [^211^At]NaAt solution, pH = 7, ~85 MBq (2.3 mCi), was added 20 μL of a 1 mg/mL aqueous solution of ChT. The mixture sat at room temperature for 5 min before conducting radio-HPLC analysis for the preparation of [^211^At]**12**. Radio-HPLC indicated that a 100% conversion was obtained. Radio-HPLC t_R_ = 12.5 min.

#### 4.5.17. *Para*-Iodoxybenzamidyl-dPEG_4_-Carboxylic Acid Methyl Ester, **11**

To a 10 mL MeOH solution containing 100 mg (0.196 mmol) of **10** was added 149 mg (0.432 mmol; 50% purity) *m*CPBA. The resultant solution was heated at 70 °C in a microwave reactor for 20 min. The conversion of **10** to **11** was followed by HPLC. The reaction mixture was evaporated to dryness using rotary evaporator under vacuum. The crude **11** was dissolved in 1:1 MeOH:H_2_O and purified using Biotage Flash Purification System (Charlottesville, VA, USA) to yield 14 mg (13.2%). ^1^H NMR (DMSO-d_6_): δ 2.23 (t, 2H, J = 6.2 Hz), 2.79 (t, 2H, J = 5.7 Hz), 3.46–3.53 (m, 15H), 3.57 (t, 2H, J = 6.2 Hz), 3.89 (s, 3H), 8.08–8.14 (m, 4H). HRMS (ES^+^): C_19_H_28_INNaO_9_ (M+Na)^+^ Calcd: 564.0707 Found: 564.0762. HPLC t_R_ = 5.0 min.

#### 4.5.18. Oxidation of [^125^I]**10** to Prepare [^125^I]**11**

To the reaction mixture containing [^125^I]**10** (Section 4.5.13. above) was added 100 μL of a 0.05 M solution of *m*CPBA in NaOH. The solution was stirred and heated at 70 °C for 20 min under microwave heating. After that time the reaction solution was analyzed by radio-HPLC to give a 60% radiochemical yield of [^125^I]**11**. Radio-HPLC t_R_ = 7.2 min.

#### 4.5.19. Oxidation of [^211^At]**12** to Prepare [^211^At]**13**

To the reaction mixture containing [^211^At]**12** (Section 4.5.14. above) was added 100 μL of a 0.05 M solution of *m*CPBA in NaOH. The solution was stirred and heated at 50 °C for 20 min under microwave heating. After that time the reaction solution was analyzed by radio-HPLC to give a 74% radiochemical yield of [^211^At]**13** by radio-HPLC analysis (t_R_ = 5.8 min).

#### 4.5.20. Preparation of [^125^I]NaIO_3_

To a 600 μL solution containing 0.01 M NaIO_4_ in 0.1 N H_2_SO_4_ was added a 2 μL solution containing 41 kBq (1.27 μCi) of [^125^I]NaI. The reaction mixture sat at room temperature for 3 days to achieve complete conversion of [^125^I]NaI to [^125^I]NaIO_3_. The conversion was followed by radio-HPLC where non-radioactive NaIO_3_ was used as a retention time reference standard [38].

(Animal study) To a 500 μL solution containing 0.025 M NaIO_4_ in 0.1 N H_2_SO_4_ was added a 1 μL solution containing (1.22 mCi) of [^125^I]NaI. The solution was stirred and heated to 120 °C for 10 min under microwave heating. The conversion was followed by radio-HPLC where non-radioactive NaIO_3_ was used as a retention time reference standard. The [^125^I]NaIO_3_ was purified by collecting the radio-HPLC peak. A 62% radiochemical yield was obtained. A portion of the isolated [^125^I]NaIO_3_ was used in the biodistribution study after the removal of HPLC solvents and dissolution in PBS.

#### 4.5.21. Preparation of [^211^At]NaAtO_3_

To a 200 μL solution of 0.01 M NaIO_4_ in 0.1 N H_2_SO_4_ (200 μL) was added 100 μL solution containing 68 MBq (1.8 mCi) of [^211^At]NaAt. Radio-HPLC analysis indicated that the [^211^At]NaAt was completely converted to [^211^At]NaAtO_3_ in less than 30 min at room temperature. The [^211^At]NaAtO_3_ was purified by collecting the radio-HPLC peak. A portion of the isolated [^211^At]NaAtO_3_ was used in the biodistribution study after the removal of HPLC solvents and dissolution in PBS.

### 4.6. Biodistribution Studies

All animal studies were approved by the Institutional Animal Care and Use Committee (IACUC) protocol 2485-05 at the University of Washington and were conducted in accordance with the NIH guidelines. Athymic mice (Crl:NU-Foxn1nu) were obtained from Charles River Laboratories (Hollister, CA, USA). In each experiment, the radioactive product was isolated from the radio-HPLC effluent, the organic solvent was removed by rotary evaporation, then the residue was dissolved in phosphate buffered saline (PBS) to prepare the doses for injection. The isolated ^125^I-labeled product was coinjected with the corresponding isolated ^211^At-labeled product to minimize the number of mice required.

In one biodistribution study, products from the reactions to make [^125^I]**11** and [^211^At]**13** were coinjected. A 100-μL volume of injectate containing ~3 μCi (~81 kBq) of each radionuclide was administered to each of 15 mice via tail vein injection. In a second biodistribution study ~5 μCi (~135 kBq) of [^125^I]NaIO_3_ and [^211^At]NaAtO_3_ were coinjected in a 100-μL dose to each mouse. Groups of 5 mice were euthanized at 1, 4 and 24 h post injection (p.i.) under ketamine/xylazine (130 mg/8.8 mg/kg) anesthesia by cervical dislocation and selected tissues were excised. Cardiac blood samples were collected under anesthesia immediately before euthanasia. Total blood weight was estimated to be 6% of the body weight. The tissues excised includee muscle, lung, kidney, spleen, liver, intestine, neck (thyroid) and stomach. Excised tissues were blotted free of blood and weighed prior to measuring the radioactivity in them.

Quantification of percent injected dose per gram (%ID/g) in the tissues was calculated by averaging the count of 5 × 1 μL samples of the injectate as standards for a total of ^125^I and ^211^At counts administered. The total quantity injected (μL) was obtained by weighing the syringe before and after injection. Radioactivity counting was conducted in a gamma counter and each sample was counted twice. The first count obtained total activity (^211^At + ^125^I). A second count of the samples was done at 4–5 days post the first count to allow all of the ^211^At to decay. Thus, the second count represented only ^125^I counts. The quantity of ^211^At in each sample was obtained by subtracting the second counts from the first count. The ^211^At counts were corrected for decay from the beginning of the gamma counting process.

## 5. Conclusions

This study did not provide a new approach to ^211^At-labeling that would result in compounds stable to in vivo deastatination. The results showed that the oxidized arylastatine compound [^211^At]**13** is too unstable to be useful in the development of new radiopharmaceuticals; however, this study provided information that will be useful for future studies. The study was based on a hypothesis that arylastatine in higher oxidation states might be stable to in vivo deastatination and might provide a new method for labeling ^211^At and radioiodine. Some [^125^I]iodoxylaryl and [^211^At]astatoxylaryl compounds were synthesized in relatively high radiochemical yields (>70%) using *m*CPBA oxidation at 50–70 °C. While the [^125^I]iodoxylaryl compound was found to be quite stable to in vivo deiodination, the [^211^At]astatoxyaryl was found to be unstable under the conditions used for isolation. The biodistribution studies revealed drastically different in vivo tissue distributions of ^125^I-labeled [^125^I]**11** and the material isolated from the reaction to produced *p*-[^211^At]astatoxylbenzoyl-dPEG-methyl ester, [^211^At]**13**. Comparison of the radio-HPLC retention times and the tissue distributions of the isolated material from the oxidation reaction to produce [^211^At]**13** with those of [^211^At]NaAtO_3_ indicated that [^211^At]**13** likely decomposed to [^211^At]AtO_3_^-^ during the isolation process.

We continue to believe that pre-oxidation of astatination compounds may provide an alternate labeling approach for stabilizing ^211^At-labeled aromatic compounds to in vivo deastatination; it is known that in vivo oxidation of arylastatine compounds can destabilize them towards deastatination, so pre-oxidized ^211^At-labeled compounds, where the At atom is already in a higher oxidation state, might be resistant to further oxidation. For example, molecules containing At in a higher oxidation state can be formed by At-bonding with two aryl rings, forming a positive charged astatonium ion or an Ar_2_-At=O compound, which potentially can add stability to the oxidized At-complex. As in vivo deastatination is a major impediment to the development of ^211^At-labeled radiopharmaceuticals, it seems that such pre-oxidized ^211^At-labeled compounds warrant investigation.

## Figures and Tables

**Figure 1 ijms-23-10655-f001:**
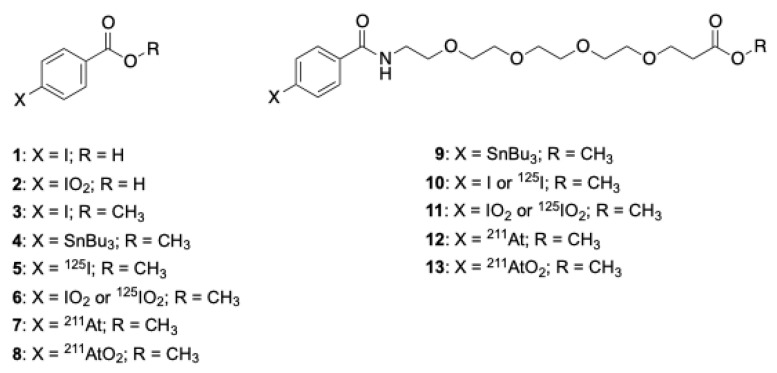
Structures of *p*-(radio)halobenzoic acid derivatives synthesized.

**Figure 2 ijms-23-10655-f002:**
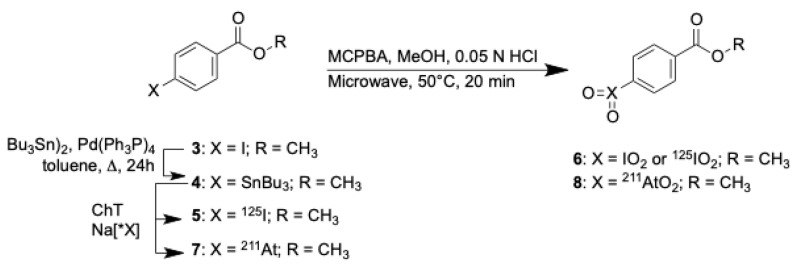
Reactions to prepare halobenzoic acid methyl esters and oxidized forms, including *p*-[^125^I]iodoxybenzoic acid methyl ester [^125^I]**6** and *p*-[^211^At]astatoxybenzoic acid methyl ester [^211^At]**8.**.

**Figure 3 ijms-23-10655-f003:**
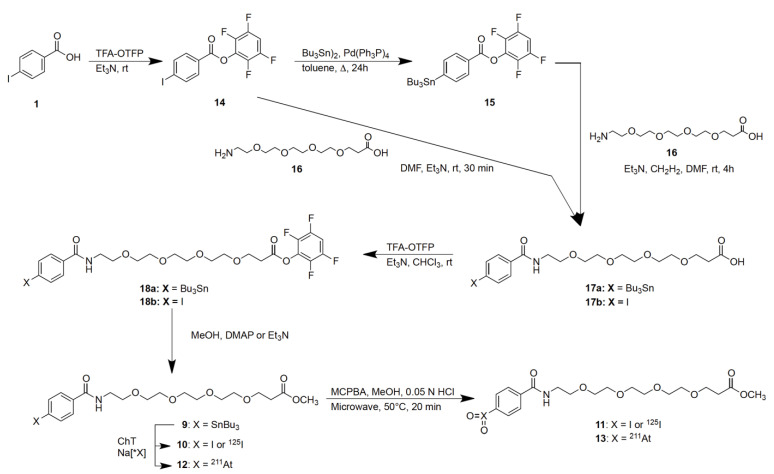
Synthetic scheme for the preparation of *p*-(radio)halobenzamidyl-dPEG_4_-carboxylic acid methyl ester derivatives **9**–**12** and subsequent oxidation to form [^125^I]**11** and [^211^At]**13**.

**Figure 4 ijms-23-10655-f004:**
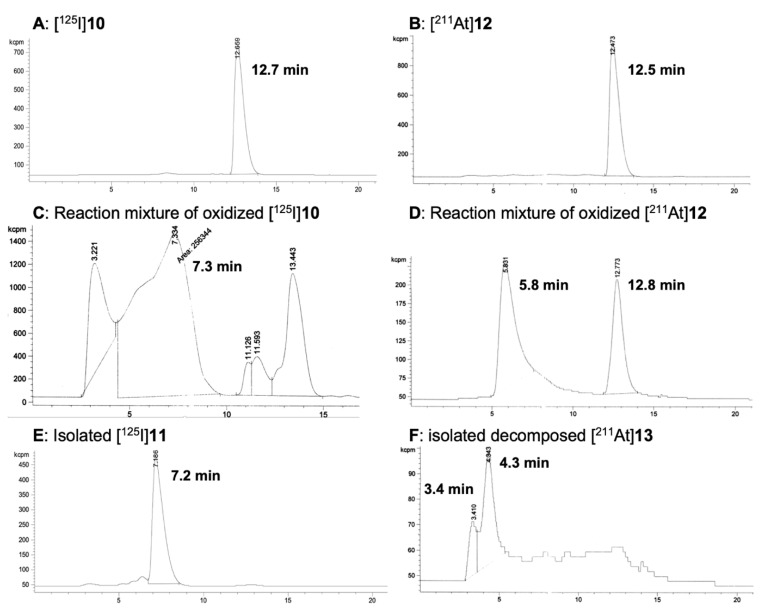
Radiochromatograms of reaction solutions and isolated products in the preparation of [^125^I]**11** and [^211^At]**13** for animal biodistributions. (Panel **A**) Reaction solution containing [^125^I]**10**; (Panel **B**) Reaction solution containing [^211^At]**12**; (Panel **C**) Reaction mixture from oxidation of [^125^I]**10**; (Panel **D**) Reaction mixture from oxidation of [^211^At]**12**; (Panel **E**) [^125^I]**11** in PBS after isolation from HPLC; (Panel **F**) Isolated products in PBS from oxidation of [^211^At]**12** (note that the desired product in panel **D** at 5.8 min decomposed on isolation).

**Figure 5 ijms-23-10655-f005:**
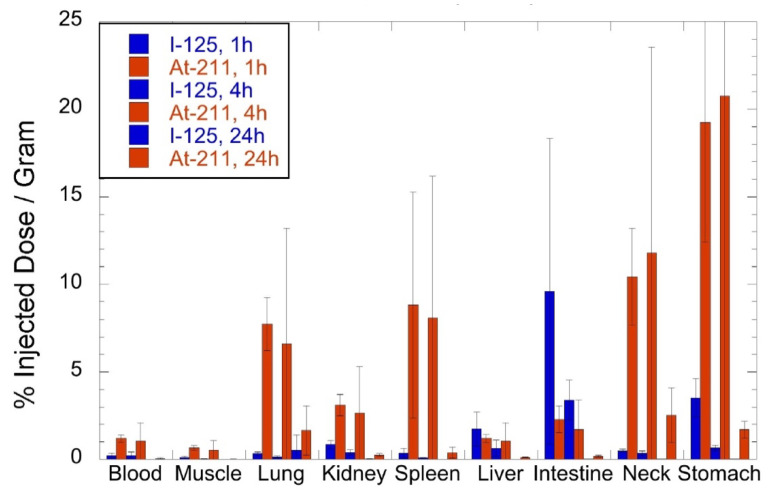
Graph showing %ID/g of isolated products for [^125^I]**11** and [^211^At]**13** at 1, 4 and 24 h p.i. in athymic mice. (n = 5) See Appendix A for a tabulation of %ID/g data.

**Figure 6 ijms-23-10655-f006:**
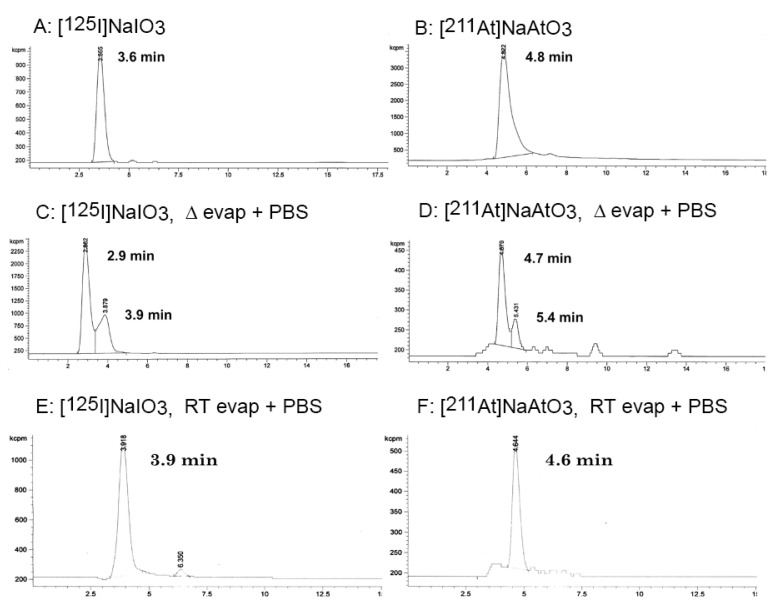
Radiochromatograms of reaction solutions and isolated products in the preparation of [^125^I]NaIO_3_ and [^211^At]NaAtO_3_ for biodistribution study. (Panel **A**) Reaction solution containing [^125^I]NaIO_3_. (Panel **B**) Reaction solution containing [^211^At]NaAtO_3_; (Panel **C**) Reaction solution containing [^125^I]NaIO_3_ after solvent evaporation under heating and dissolution in PBS; (Panel **D**) Reaction solution containing [^211^At]NaAtO_3_ after solvent evaporation under heating and dissolution in PBS; (Panel **E**) Reaction solution containing [^125^I]NaIO_3_ after solvent evaporation at room temperature and dissolution in PBS; and (Panel **F**) Reaction solution containing [^211^At]NaAtO_3_ after solvent evaporation at room temperature and dissolution in PBS. Note that retention times can slightly change when PBS solutions are injected on radio-HPLC.

**Figure 7 ijms-23-10655-f007:**
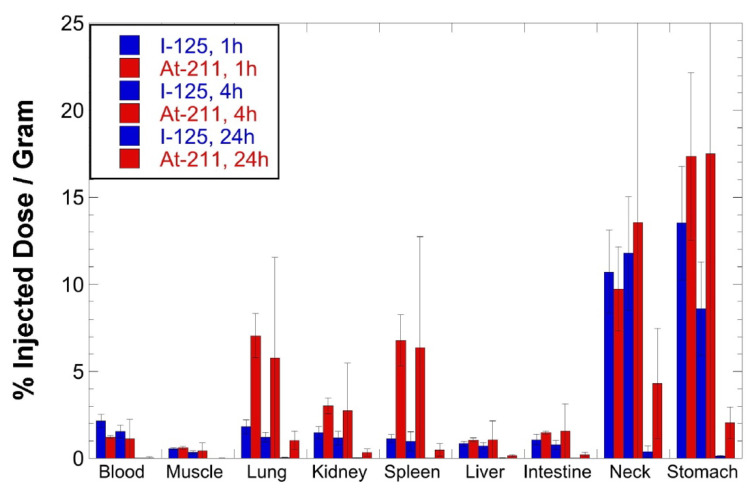
Graph showing concentration as %ID/g of [^125^I]NaIO_3_^–^ and [^211^At]NaAtO_3_^–^ in tissues of athymic mice at 1, 4 and 24 h p.i. (n = 5). See Appendix A for a tabulation of %ID/g data.

## Data Availability

The data presented in this study are available in the Appendix A (listed above).

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
