# Peer review of "Oxidation of p-[125I]Iodobenzoic Acid and p-[211At]Astatobenzoic Acid Derivatives and Evaluation In Vivo"

_ijms, 2022, doi:10.3390/ijms231810655_

Round 1

Reviewer 1 Report

* 13-14; the rational of the study/manuscript is here, unfortunately formulated  (labelling method has no link with in vivo stability of the radiopharmaceutical !), a better formulation was written in section 229-230,

* 161-162; iodate made from iodide and periodate in acid conditions - in literature you can find that this mixture gives tri-iodide !, (despite, you have verified this ?), do you have additional literature-references of this ?

* Figure 4, Pag. 5, 'F : isolated oxidized [211At]12 - should this be 13 ?

* 327 - 328 : you can add, the flow rate (of this typical run)

* biodistribution; for me, why the uptake in thyroid has not been taken up ?

Reviewer 2 Report

This paper illustrates a useful example of the application of the scientific method to the search for a solution of the problem of in vivo deastatination of At-211 radiopharmaceuticals. Authors started from an initial hypothesis that the release of At-211 occurs through a mechanism where At-211 undergoes oxidative elimination. To counteract this effect, authors speculated that the preliminary oxidation of At-211 may increase the in vivo stability of At-211-labeled compounds by preventing further enzymatic oxidation of the radioisotope. To test this hypothesis, authors carried out the oxidation of arylastatine compounds and evaluated their in vivo stability also using the corresponding I-125 analogs as a reference. Unfortunately, results were not so encouraging, but honestly authors reported and discussed their experimental observations without attempting to minimize the problems and overemphasize the potential utility of the alpha emitter At-211 as, regretfully, sometime occurs with other alpha emitters. However, it is worthy to note that also a negative result is always of high relevance for science and this study describes exactly a nice piece of radiopharmaceutical chemistry as applied to a chemically very elusive radioisotope.

Reviewer 3 Report

The idea that prompted this work was that oxidized astatine derivatives could be stable in vivo because they could not be further oxidized sounds rather intriguing in view of the earlier observation by Tézé et al. that oxidation may be a leading cause of the in vivo instability of astatine-labeled compounds. Some chemistry and biodistribution studies were performed, but because the expected oxidized products could not be synthesized, these studies did not reach any conclusion.

Introduction

Astatine-211 is indeed a radionuclide of interest for targeted radionuclide therapy. In vivo stability of the astatine-carbon bond has often been a real issue in the development of astatine-labeled radiopharmaceuticals. The discrepancy between in vitro stability, which appears reasonable considering the short half-life of astatine-211, and the in vivo behavior of several labeled compounds as well as the large differences of stability of astatine labeled proteins, such as full IgG and antibody Fab fragments, have questioned the search for the cause of instability on the basis of chemical bond stability. However, alternative explanations have remained elusive. Enzymatic dehalogenation cannot explain experimental results. Five years ago, the possibility that astatine release could be the result of oxidation of astatine-labeled products has been investigated and the much higher sensitivity of astatine-labeled, as compared to iodine-labeled compounds, has been demonstrated by Tézé et al. It is surprising to find this paper mentioned and cited only in the discussion, and not earlier as part of the state of the art.

Results

Different chemical syntheses are described in the manuscript. However, if the chemical structure of the oxidized iodine compounds is established, those of the corresponding astatine compounds are not. An unknown product is formed that is transformed into another unknown product upon transfer into aqueous solvents. The similarity of the biodistribution of this unknown astatine compound with that of oxidized astatine is considered an indication that the oxidized astato-benzoate has been spontaneously degraded into free, oxidized astatine, but no chemical characterization is provided. Since the most stable isotope of astatine has a half-life not much longer than 8 hours, direct chemical characterization of astatine derivatives is indeed difficult. Besides, solvent evaporation under heating or at room temperature did not afford the same products, according to HPLC and this remained unexplained.

In addition, it is stated that "Oxidation of [211At]NaAt provided a new species, presumably [211At]NaAtO3". Later-on, the "presumably" disappears and reference is made to [211At]NaAtO3. This point should be clarified since [211At]AtO3-is not mentioned in the Pourbaix diagram of astatine by Sergentu et al. Advances on the Determination of the Astatine Pourbaix Diagram: Predomination of AtO(OH)2- over At- in Basic Conditions. Chem. Eur. J. 2016, 22, 2964 – 2971.

Altogether, study of the biodistribution of the proposed oxidized astato-benzoate could not be performed because the product could not be synthesized. The biodistribution of the non-oxidized compounds was not described.

Discussion

It is stated that "The primary goal of this study was to evaluate whether a specific type of astatinated aryl compound that has the At atom in a higher oxidation state might be stable to in vivo deastatination". Since such compounds could not be synthesized, it is not clear that any conclusion can be made other than a compound entirely unstable in vitro will probably not be more stable in vivo.

Round 2

Reviewer 3 Report

The reviewer is fully aware of the difficulty of working with an elusive, short-lived, radionuclide such as astatine-211 and of the lack of a stable isotope.

The reviewer maintains that the authors "did not obtain the 211At-labeled compound [the authors] stated was obtained and the belief that [the authors] did not fully characterize the 211At-labeled compound". This is not a mere "impression", but a straightfroward conclusion based on the authors statements. The abstract summarizes the findings by stating that "The oxidized [211At]benzoic acid derivatives appeared unstable under the conditions of isolation" and "Evaluation of the oxidized [211At]benzamide injectate by radio-HPLC (post animal injection) indicated that it had changed during the isolation process. Comparison of the biodistributions of the oxidized [211At]benzamide derivative and of [211At]astatate confirmed that species obtained after isolation was likely [211At]astatate". For the reviewer, obtaining a compound involves not only a chemical synthesis but also the isolation of the compound in a form that allows some kind of characterization. In addition, for an in vivo study to support any conclusion about the in vivo stability or instability of a chemical compound, the compound must at least be administered to a living organism. The authors make it quite clear that this was not the case and that the in vivo studies were most likely performed using [211At]astatate.

Round 3

Reviewer 3 Report

The authors have now clearly expressed that oxidizing astato-benzoate only affords astate and that the proposed approach cannot be used for astatine targeting.